# Topology Optimization of Turbulent Flow Cooling Structures Based on the k-ε Model

**DOI:** 10.3390/e25091299

**Published:** 2023-09-05

**Authors:** Yiwei Sun, Menglong Hao, Zexu Wang

**Affiliations:** 1School of Materials Science and Engineering, Southeast University, Nanjing 210096, China; sunyiwei@seu.edu.cn; 2Key Laboratory of Energy Thermal Conversion and Control of Ministry of Education, School of Energy and Environment, Southeast University, Nanjing 210096, China; haom@seu.edu.cn

**Keywords:** k-ε turbulent model, fluid topology optimization, fluid–solid conjugate heat transfer, variable density method, cooling channel design

## Abstract

Topology optimization (TO) is an effective approach to designing novel and efficient heat transfer devices. However, the TO of conjugate heat transfer has been essentially limited to laminar flow conditions only. The present study proposes a framework for TO involving turbulent conjugate heat transfer based on the variable density method. Different from the commonly used and oversimplified Darcy model, this approach is based on the more accurate and widely accepted k-ε model to optimize turbulent flow channels. We add penalty terms to the Navier–Stokes equation, turbulent kinetic energy equation, and turbulent energy dissipation equation, and use interpolation models for the thermal properties of materials. A multi-objective optimization function, aiming to minimize the pressure drop and the average temperature, is set up to balance the thermal and hydraulic performance. A case study is conducted to compare various optimization methods in the turbulent regime, and the results show that the present method has substantially higher optimization effectiveness while remaining computationally inexpensive.

## 1. Introduction

Many engineered products are subject to various degrees of thermal loads during operation and often must be actively cooled with a fluid. For many of these products, their performance relies heavily on cooling effectiveness. For example, gas turbine vanes and blades draw cold air to flow through their internal channels, thus allowing higher working temperatures and boosting power and efficiency [1,2]. Electronic devices continue to be miniaturized and are generating increasingly more waste heat that is difficult to remove, which has become a bottleneck for future device development [3,4,5]. Lithium-ion batteries also produce large amounts of heat during very fast charging and accidental thermal runaways and require more effective liquid cooling than is currently available [6,7,8]. 

The design of the cooling channels has a significant impact on the cooling performance, and the goal of cooling channel design is typically to minimize the flow resistance while maximizing the convective heat transfer rate [9,10,11,12]. Conventional fluid channel design practice is largely based on intuition and a trial-and-error process, which is both inefficient and ineffective in generating the best structure. Topology optimization as an algorithm-based design method has been successfully applied to the optimization of mechanical structures since it was proposed by Bendsøe and Kikuchi [13]. Popular topology optimization methods include homogenization methods [13], evolutionary structural optimization (ESO) methods [14], variable density methods [15], level set methods [16], moving morphable components (MMC) methods [17], etc. As for topology optimization of fluid flow, Borrvall and Petersson [18] pioneered a method of adding a penalty term to the Stokes equation to penalize the fluid velocity and realize the optimal design of flow channels. Among the following studies, A. Gersborg-Hansen et al. [19] extended the fluid topology optimization to Navier–Stokes flow with low Reynolds numbers based on Borrvall’s research. Pietropaoli M et al. [20] proposed a design method based on topology optimization for the internal cooling channels of the gas turbine. Han et al. [21] used topology optimization to design spider-web-shaped heat sinks and experimented with different objective functions. The resulting structure has a 57.35% lower spatial temperature difference (*T_max_*−*T_min_*) and 8% lower thermal resistance compared with the traditional design. Chen et al. [22] designed a topology-optimized cold plate for a lithium-ion battery thermal management system and achieved a higher heat transfer coefficient and significantly lower spatial temperature difference. Hu et al. [23] applied topology optimization to design microchannel heat sinks and demonstrated superior cooling performance compared with a conventional design of straight fin microchannel. 

Turbulent flow is ubiquitous in advanced cooling structures, such as the active cooling of gas turbine blades, the regenerative cooling of rocket engines, and even the liquid cooling of high-power density electronics. However, topology optimization of conjugate heat transfer involving turbulent flow has been challenging because of the combined strong nonlinearity of turbulence, thermal–fluid coupling, and topology optimization itself. For pure turbulent fluid flow problems, a number of topology optimization schemes have been developed. Yoon [24] proposed a topology optimization method based on the Spalart–Allmaras (SA) turbulence model using the finite element method. Luís F.N. Sáet al. [25] extended the Spalart–Allmaras turbulence topology optimization method to flow channel design in a rotating domain. Yoon [26] developed a new topology optimization method based on the k-ε turbulent model by adding penalty terms to the Navier–Stokes equation, the turbulent kinetic energy equation, and the turbulent energy dissipation equation, respectively. Dilgen et al. [27] proposed a topology optimization method applicable to the k-ω turbulence model and the topology optimization flow channel design significantly reduced the vortex. For conjugate heat transfer design problems in the turbulent regime, the development of topology optimization methods is still at a much earlier stage. The majority of existing studies adopt the Darcy flow model to achieve stable numerical convergence and reduce computation time. Zhao et al. [28] used a low-cost Darcy flow model as an approximate substitute for turbulence flow in a topology optimization problem considering conjugate heat transfer. An-Li [29] studied the method of permeability setting and pressure drop constraint setting based on the Darcy flow topology optimization. In the topology optimization of the two-dimensional flow channel, Li et al. [30] used a simplified Darcy flow model to approximate turbulent flow. They applied the convective heat transfer model and Darcy flow model to solve the microchannel region’s temperature and pressure fields, and the microchannel layout in a heat sink is designed by topology optimization. While the computational cost is low, these Darcy-based methods do not capture the essence of turbulence due to a lack of inertia terms, and therefore greatly sacrifice the optimization quality. Dilgen et al. [31] presented a novel topology optimization method based on the k-ω turbulence model. However, the objective function did not consider flow resistance, which must be taken into account in typical cooling designs. In addition, the k-ω model is mainly applied to fluid flow at a low Reynolds number and shear flow, and the applicability is limited. 

The k-ε turbulent RANS model is one of the most effective and widely used turbulent flow models in engineering. However, an effective approach is lacking currently to apply the k-ε model in topology optimization for conjugate heat transfer. In this study, a conjugate heat transfer topology optimization method based on the k-ε turbulent model is developed to design cooling channels. The variable density method is used for optimization. The novelty is mainly reflected in the penalty terms which are added to the momentum equation, turbulent kinetic energy equation, and dissipation rate equation, respectively, and the material’s thermal properties are interpolated to distinguish the solid and fluid phases. Then, the turbulent conjugate heat transfer topology optimization equations are obtained. The minimum inlet and outlet pressure drop and the minimum average temperature are set as multi-objective functions to meet the requirement of hydraulic and thermal performance. The turbulent conjugate heat transfer topology optimization structure is developed by carrying out the sensitivity analysis with the MMA algorithm in the topology optimization process. The optimization results using the k-ε model are compared with those using the Darcy flow model as well as the conventional straight flow channel, and the topology optimization method based on the k-ε turbulent model showed the lowest flow resistance and average temperature in the design domain.

## 2. Model of the Topology Optimization Problem

In this paper, the k-ε turbulent model and the conjugate heat transfer model are used to realize the topology optimization design of cooling channels. The turbulent conjugate heat transfer topology optimization conceptual design model is shown in Figure 1. The physical design domain consists of the solid phase domain Ω_s_ and the fluid domain Ω_f_. Materials in the design domain are regarded as porous media, and the design variable γ, which affects the material’s physical properties, varies from zero to unity, where γ = 1 represents fluid and γ = 0 represents solid. The solid isotropic material with penalization (SIMP) interpolation model is used to speed up the conversion of porous media to solids or fluids during the iterative process. A uniform heat input *Q* is applied to the entire design domain and different boundary conditions are set, including inlet *Γ_in_*, outlet *Γ_out_*, and wall boundaries *Γ_wall_* according to the actual flow and heat transfer conditions. 

### 2.1. Governing Equations

The pseudo-variable γ is introduced into the flow and heat transfer control equations based on the variable density method. The flow and heat transfer equations are solved to compute the temperature, velocity, and pressure for the entire design domain first, and then the pseudo-variables are iteratively updated through sensitivity analysis and optimization solutions. Subsequently, the updated pseudo-variables are used to optimize the temperature and velocity–pressure distributions in the design domain, which enables the coupling of topology optimization and the turbulent conjugate heat transfer model.

#### 2.1.1. Turbulent Flow Model

The widely used two-equation k-ε RANS model is directly implemented in the topology optimization process. The Reynolds stress tensor is regarded as a function of the turbulent viscosity in the standard k-ε turbulent model. The turbulent viscosity is solved by the turbulent kinetic energy equation (k-equation) and turbulent energy dissipation equation (ε-equation).

In laminar flow topology optimization, a penalty term f related to the flow velocity is added to the momentum equation to distinguish solid and fluid regions in the design domain. In the fluid region, the penalty term is approximately equal to 0, while in the solid region, the penalty term is infinitely large, and the flow velocity tends to reach 0. The penalty term f is obtained by interpolation of the design variables. This method is also known as the Brinkman penalty model, and it is currently the most common method for solving laminar flow topology optimization.

As for solving topology optimization involving turbulent flow using the k-ε turbulence model, in addition to the penalty term in the momentum equation, the influence of the turbulent kinetic energy and the turbulent energy dissipation rate on the topology optimization design needs to be considered. Therefore, additional penalty terms $f_{k}$ and $f_{\varepsilon }$ are added to the turbulent kinetic energy equation and turbulent energy dissipation equation, respectively. The fluid has turbulent kinetic energy and energy dissipation, so $f_{k}$ and $f_{\varepsilon }$ tend to be zero in the fluid region. Whereas there is no turbulent kinetic energy and energy dissipation in the solid region, so $f_{k}$ and $f_{\varepsilon }$ tend to be infinity in solid region.

For incompressible steady-state fluid, the conservation equations for turbulent topology optimization are given as follows:(1)∇⋅u=0
(2)$$\rho \left(u\cdot \nabla \right)u=\nabla \cdot \left[-pI+\left(\mu +\mu_{T}\right)\left(\nabla u+\nabla u^{T}\right)\right]+f,f=-\alpha_{f}\left(\gamma \right)u$$

The modified (added penalty terms $f_{k}$ and $f_{\varepsilon }$) turbulent kinetic energy equation and energy dissipation equation are expressed as:(3)$$\rho \left(u\cdot \nabla \right)k=\nabla \cdot \left[\left(\mu +\frac{\mu_{T}}{\sigma_{k}}\right)\nabla k\right]+P_{k}-\rho \varepsilon +f_{k},f_{k}=-\alpha_{k}\left(\gamma \right)k$$
(4)$$\rho \left(u\cdot \nabla \right)\varepsilon =\nabla \cdot \left[\left(\mu +\frac{\mu_{T}}{\sigma_{\varepsilon }}\right)\nabla \varepsilon \right]+C_{1\varepsilon }\frac{\varepsilon }{k}P_{k}-C_{2\varepsilon }\rho \frac{\varepsilon^{2}}{k}+f_{\varepsilon },f_{\varepsilon }=-\alpha_{\varepsilon }\left(\gamma \right)\varepsilon$$
where
(5)$$\mu_{T}=\rho C_{\mu }\frac{k^{2}}{\varepsilon }$$
(6)$$P_{k}=\mu_{T}\left[\nabla u:\left(\nabla u+{\left(\nabla u\right)}^{T}\right)\right]$$

In the turbulent kinetic energy equation (Equation (3)) and the energy dissipation equation (Equation (4)), $\mu_{T}$ is the turbulent viscosity and $P_{k}$ is the production term. The parameters in Equations (3) and (4) recommended by the most literature are $\sigma_{k}=1$, $C_{1\varepsilon }=1.44$, $C_{2\varepsilon }=1.92$, $\sigma_{\varepsilon }=1.3$, $C_{\mu }=0.09$. Penalty terms f, $f_{k}$, and $f_{\varepsilon }$ are applied to the momentum equation, the turbulent kinetic energy equation, and the energy dissipation equation, respectively. $\alpha_{f}$, $\alpha_{k}$, $\alpha_{\varepsilon }$ are the local inverse permeability of the porous medium, which can be calculated by the following interpolation functions.
(7)$$\alpha_{f}\left(\gamma \right)=\alpha_{fmax}+\left(\alpha_{fmin}-\alpha_{fmax}\right)\gamma \frac{1+p_{\alpha }}{\gamma +p_{\alpha }}$$
(8)$$\alpha_{k}\left(\gamma \right)=\alpha_{kmax}+\left(\alpha_{kmin}-\alpha_{kmax}\right)\gamma \frac{1+p_{\alpha }}{\gamma +p_{\alpha }}$$
(9)$$\alpha_{\varepsilon }\left(\gamma \right)=\alpha_{\varepsilon max}+\left(\alpha_{\varepsilon min}-\alpha_{\varepsilon max}\right)\gamma \frac{1+p_{\alpha }}{\gamma +p_{\alpha }}$$
where γ is the design variable and $p_{\alpha }$ is the resistance coefficient penalty factor which is set to 0.01. $\alpha_{fmin}$, $\alpha_{kmin}$, and $\alpha_{\varepsilon min}$ are the inverse permeability in the fluid region and are equal to 0 in this paper. $\alpha_{fmax}$, $\alpha_{kmax}$, and $\alpha_{\varepsilon max}$ are the inverse permeability in the solid region and they should be given as infinity theoretically. However, limited by the computer’s storage space, the values of $\alpha_{fmax}$, $\alpha_{kmax}$, $\alpha_{\varepsilon max}$ are set as 10^7^, 10^8^, and 10^8^, respectively.

#### 2.1.2. Conjugate Heat Transfer Model

The SIMP interpolation model is adopted for the thermal properties of materials.
(10)$$k\left(\gamma \right)=\left(k_{f}-k_{s}\right)\gamma^{p_{1}}+k_{s}$$
(11)$$\rho \left(\gamma \right)=\left(\rho_{f}-\rho_{s}\right)\gamma^{p_{2}}+\rho_{s}$$
(12)$$c\left(\gamma \right)=\left(c_{f}-c_{s}\right)\gamma^{p_{3}}+c_{s}$$
where k, ρ, c are the thermal conductivity, density, and specific heat capacity of the materials. The physical parameters of the solid and fluid materials are shown in Table 1. In order to achieve the 0–1 structure, it is necessary to introduce a penalty factor *p* to penalize the intermediate variables so as to suppress the influence of the intermediate variables on the results. The value of the penalty factor needs to be selected according to actual problems and numerical results. In this paper, the penalty factors *p*_1_, *p*_2_, *p*_3_ are set to 3, 3, 1.

The conjugate heat transfer equation can be given as
(13)ρ(γ)c(γ)(u⋅∇T)=∇⋅(k(γ)∇T)+Q

### 2.2. Topology Optimization Model

When designing cooling channels, both hydraulic and thermal performance need to be considered, which can be quantitatively evaluated by the pressure drop and average temperature, respectively. In this paper, the optimization objectives are set to reduce the pressure drop of the fluid from the inlet to the outlet as well as lower the average temperature in the design domain. The pressure drop and average temperature can be calculated by the following equations, respectively.
(14)$$\psi =\frac{1}{\left|\Gamma_{2}\right|}\int_{\Gamma_{2}}pd\Gamma -\frac{1}{\left|\Gamma_{1}\right|}\int_{\Gamma_{1}}pd\Gamma$$
(15)$$\phi =\frac{1}{\left|\Omega \right|}\int_{\Omega }Td\Omega$$
where *Γ*_2_ and Γ_1_ are the inlet and outlet boundaries, and the Ω is the design domain. The hydraulic and thermal objectives are normalized to avoid numerical instabilities in the solution process. Then, objective functions are weighted and added to obtain the multi-objective function, which is given as follows:(16)$$\Pi =w_{i}\frac{\phi }{\phi_{0}}+w_{j}\frac{\psi }{\psi_{0}}$$
(17)$$w_{i}+w_{j}=1$$
where $w_{i}$ and $w_{j}$ are the weighting factors of the multi-objective function. $\psi_{0}$ and $\phi_{0}$ are the initial values of pressure drop and average temperature, respectively.

The topology optimization problem can be described as
(18)$$Find\;\;\;\;\;\gamma_{i}\left(i=1,2,\ldots ,N\right)\;Minimize\;\Pi =w_{i}\frac{\phi }{\phi_{0}}+w_{j}\frac{\psi }{\psi_{0}}\;Subject\;to\{\begin{array}{c} \int_{\Omega }\gamma d\Omega ⩽v_{f}\int_{\Omega }1d\Omega \\ 0⩽\gamma_{i}⩽1\left(i=1,2,\ldots ,N\right) \end{array}$$
where $v_{f}$ is the volume fraction of fluid and is set to 0.5. $\gamma_{i}$ is the design variable in each domain element. The temperature field, pressure field, and velocity field are discretized using Lagrange linear interpolation. The GCMMA algorithm is used for topology optimization [23].

### 2.3. Density Filtering

Checkerboard is a common problem in topology optimization results, which manifests as a periodic distribution of solid and fluid elements and is thought to be caused by poor numerical modeling [32]. The appearance of the checkerboard is usually inhibited by density filtering. Moreover, it should be noted that for the conjugate heat transfer problem, density filtering can reduce the ill-posedness of the topology optimization results [21]. A density filter based on the Helmholtz equation is used to avoid checkerboard and mesh-dependency problems in topology optimization [33] and is defined as follows:(19)$$-r^{2}\nabla^{2}\tilde{\gamma }+\tilde{\gamma }=\gamma$$
where, $\tilde{\gamma }$ is the filtered design variable and r is the filter radius.

### 2.4. Numerical Example

The numerical example of turbulent conjugate heat transfer topology optimization is shown in Figure 2. Due to the symmetry of the domain, the upper half of the region is selected as the design domain and a symmetrical boundary condition is imposed on the symmetrical line. The characteristic length of the model L is 0.02 m. A Reynolds number of 10,000 ($Re=\frac{\rho_{in}U_{in}L}{\mu_{in}}=\frac{1000\times 0.5\times 0.02}{0.001}=10,000$) is used in this example. For the temperature field, the inlet temperature is prescribed (*T_in_* = 293.15 K), and the other boundary conditions are adiabatic ($-\vec{n}\cdot q=0$). For the velocity and pressure field, the velocity magnitude, turbulent kinetic energy, and turbulent energy dissipation at the inlet are given ($U_{in}=0.5\;m/s$, $k_{0}=0.005\;m$^2^/s^2^, $\varepsilon_{0}=0.005\;m$^2^/s^3^), and the pressure at the outlet is set as $p_{out}=0$. The wall condition is non-slip, and the turbulent flow in the near-wall region is simulated by the wall function method. The design domain is discretized into 13,427 four-node quadratic elements and the initial value of the design variable γ is set to 0.5.

The hydraulic and heat transfer performance of topology optimization channels is compared with the traditional straight channel in Section 3.2. The traditional straight channel structure is shown in Figure 3, where the length and width of the fins are 80 mm and 7.8125 mm, respectively. The volume fraction of the fins is the same as the solid region of topology optimization models.

## 3. Results and Discussion

### 3.1. Comparison of Several Conjugate Heat Transfer Topology Optimization Designs

We carry out topology optimization using three different methods, namely the Navier–Stokes flow model, the Darcy flow model, and our k-ε-based model, under the same problem settings, and compare their outcomes. In the Navier–Stokes flow model, the penalty term is only added to the momentum equation but not to the turbulent kinetic energy equation and turbulent energy dissipation equation. Convergence histories of the objective function for all three methods are shown in Figure 4, and the total number of iterations and the computation time are summarized in Table 2. As can be seen in these results, the Navier–Stokes flow model requires the fewest iterative steps, but each step is time-consuming, and the total computing time is the longest. Using the Darcy flow model largely reduces the computation time. However, its final objective function value is larger than that of other flow models. When the k-ε turbulent flow model is used, the computation time is longer than using the Darcy model but still within an acceptable range. The penalty terms imposed on the turbulent kinetic energy equation and turbulent energy dissipation equation accelerate the transition of the design variables to the solid or liquid phase in the topology optimization process. Moreover, the k-ε model yields the lowest objective function value thanks to the additional penalty terms. Considering both the optimization efficiency and the final optimization object function, the k-ε model yields the best optimization results among the 3 models used.

The evolution of design variables and the velocity distribution during the optimization process are shown in Figure 5. There are only two flow channels in the design domain in the Navier–Stokes flow model. Such a simple flow channel configuration cannot effectively improve the heat transfer performance of the conjugated heat transfer structure. When the Darcy flow model is used, the shape of the solid region looks like multiple fins. The fluid is evenly distributed by the fins at the inlet and flows into different flow channels which are nearly parallel. In the k-ε turbulent flow model, the solids in the design domain are mostly distributed in blocks, so the flow channel has more branches to enhance the heat transfer effect between the fluid and the solid.

### 3.2. Validation of Topology Optimization Results

The k-ε-optimized flow channel and Darcy-optimized flow channel are compared with the traditional straight flow channel. A uniform heat source (*Q* = 10^7^ W/m^2^) is applied to the design domain, and the boundary conditions are the same across different models. The k-ε turbulent flow physical model and conjugate heat transfer physical model are used to evaluate the hydraulic and heat transfer performance of the resulting designs. 

#### 3.2.1. Grid Independence Test

Firstly, a grid independence test is conducted, and the k-ε-optimized channel is used for the test. The model is considered grid independent when the deviation of the numerical results is within 2%. The test results are shown in Table 3. It can be concluded that when the number of the grid is about 300,000, the balance between calculation accuracy and computing time can be satisfied. Therefore, Grid 2 is used for the k-ε-optimized flow channel generated. The number of grids in the Darcy-optimized flow channel is 382,437, and that in the traditional straight channel is 330,378.

#### 3.2.2. Flow Characteristics Analysis

Figure 6 shows the pressure drop of the different types of flow channels with different inlet velocities. It is clear that, regardless of the design of the flow channel, the pressure drop increases with increasing inlet velocity. With the same inlet velocity, the pressure drop of the topology-optimized flow channel is obviously lower than that of the straight flow channel. Moreover, the k-ε-optimized flow channel has a lower pressure drop than the Darcy-optimized flow channel when the inlet velocity is the same. The higher the inlet velocity, the larger the difference in pressure drop between the three types of channels. With the highest inlet velocity of 1.5 m/s, the pressure drop of the straight flow channel is 2760 Pa, while the pressure drop is only 1110 Pa for the Darcy design and 837 Pa for the k-ε design. Therefore, the flow channel designed by topology optimization can significantly decrease the pressure drop and the energy dissipation during fluid flow.

The relative difference in pressure drop *δ* is used to demonstrate the hydraulic performance improved by different types of topology-optimized flow channels compared with the conventional channel, as is shown in Figure 6. *δ* is defined as:(20)$$\delta =\frac{\Delta P_{Str}-\Delta P_{To}}{\Delta P_{Str}}$$
where the $\Delta P_{Str}$ and $\Delta P_{To}$ are the pressure drop of the conventional straight flow channel and topology-optimized flow channel, respectively. It can be seen that using the Darcy-optimized flow channel reduced pressure drop by 31.79–59.79%, depending on the inlet velocity, compared with the straight flow channel. Using the k-ε-optimized flow channel reduced the pressure drop more effectively by 56.13–69.68%. Notably, although the inlet velocity selected for the topology optimization design is 0.5 m/s, the k-ε-optimized flow channel has a better hydraulic performance at all inlet velocities, and especially at higher inlet velocities. Therefore, it can be considered that topology-optimized designs using the k-ε turbulent model have applicability beyond their design condition.

Figure 7 shows the streamline profile at a low inlet fluid velocity (0.5 m/s) and a high inlet velocity (1.5 m/s) in different types of flow channels. For the straight flow channel, vortices with different sizes can be found in the inlet and outlet manifolds of the parallel flow channels. This flow pattern indicates significant pressure loss and energy dissipation in the fluid flow process whatever the inlet velocity. For the Darcy-optimized flow channel, in the case of low inlet velocity, it appears that the abrupt expansion between the inlet and the heat transfer region generates large vortexes and hinders the fluid flow, which indicates higher local energy loss and an increase in the overall pressure drop. When the inlet velocity is 1.5 m/s, the fluid is concentrated in the centerline of the structure. When the fluid flows near the outlet, the sudden shrinkage of the channel also causes vortices and some local pressure loss. In addition, when the high-speed fluid hits the outlet wall, it flows back toward the inlet along the uppermost channel, which obviously hinders effective heat exchange. As for the k-ε-optimized channel design, it can be seen that the bends with fillets are formed at the bifurcation and confluence of channels. The flow is slightly disturbed, and the velocity distribution is uniform and stable. Only a few small vortices are generated when the velocity increases. These streamline results very well explain the pressure drop differences in Figure 6. The k-ε-optimized flow channel needs a lower pressure drop under the same inlet velocity largely due to less energy dissipation in vortices.

#### 3.2.3. Heat Transfer Characteristics Analysis

The heat transfer performance of different flow channels is evaluated by comparing the temperatures of the designed structure. Figure 8 shows the average temperature and maximum temperatures with different inlet velocities for the three cooling structures. For the k-ε-optimized flow channel and the straight flow channel, the average temperature and maximum temperature decrease as the inlet velocity increases. In contrast, the temperature in the Darcy-optimized channel initially rises and then decreases with an increasing inlet velocity, possibly due to the occurrence of the reversed flow in the uppermost channel as mentioned above. When the inlet velocity is lower than 0.5 m/s, the coolant mainly flows through the channels that are close to the inlet, so the heat transfer of the flow channels farther away from the inlet is poor, and local hot spots are generated. This disadvantage is exacerbated at the velocity of 0.5 m/s. Generically, when the inlet flow velocity increases, the convection between the fluid and the solid is enhanced, so the average temperature and the maximum temperature both decrease, although at the cost of a higher pressure drop. The average temperature and maximum temperature of the conjugate heat transfer structure with the k-ε-optimized flow channel are lower than those of the Darcy-optimized flow channel and straight flow channel under the same inlet flow rate. The maximum temperature of the Darcy flow channel is as high as 485 K and reduces to 336 K at the highest flow rate. In comparison, the maximum temperature stays below 328 K for the design by the k-ε model, indicating a much better cooling performance in terms of eliminating hot spots.

Figure 9 shows the spatial temperature distribution of the structure when the inlet velocity is 0.5 m/s and 1.5 m/s, respectively. Compared with the other two channel configurations, the temperature distribution of the Darcy-derived design is highly non-uniform showing high local temperature gradient. This can be explained by the flow patterns with large vortices and reversed flow as shown in Figure 7. For the k-ε-optimized channel design, the fluid flow has better attachment with the fin in the heat transfer region, and there are no apparent hot spots. 

### 3.3. Weighting Factor Ratio w_i_:w_j_ Analysis

Decreasing the pressure drop requires the fluid to reduce friction and collision with the solid region as much as possible during the flow while improving the heat transfer performance means increasing the contact surface between the fin and the fluid, which will lead to the increase in the flow resistance and degradation of hydraulic performance. Hence, it is difficult to improve heat transfer performance and hydraulic performance simultaneously. In this section, the influence of the weighting factor ratio of the heat transfer and flow objective functions on the topology optimization design is discussed.

Figure 10 shows the topology optimization results, the streamline contour, and temperature distribution using different weight factor ratios. It can be seen that with the increase in the weight factor of the heat transfer objective function *w_i_*, the number of the branches and fins of the design increases, and the flow channels become narrower. With the same solid volume, the total length of the convective heat transfer boundary is 665.08 mm, 726.79 mm, and 904.72 mm in the cases of 1:4, 1:2, 1:1, respectively. Therefore, increasing the heat transfer weight factor extends the heat transfer boundary between the fluid and the fin, thereby enhancing the heat transfer performance. Figure 11 presents the pressure drop and the average temperature when different weighting factors are used. With the increase in the weight factor ratio, the average temperature of the heat transfer region gradually decreases. However, this better heat transfer performance is obtained at the expense of a higher pressure drop and pumping power consumption.

### 3.4. Three-Dimensional Flow Channel Model Analysis

Three-dimensional models have more degrees of freedom, so the topology optimization for three-dimensional structures needs drastically more computation time. At the same time, the complex 3D topology optimization designs will significantly increase the difficulty of manufacturing. In this section, the 2D topology optimization design is extruded into a 3D model, and the hydraulic and heat transfer performance of the 3D extruded structure in the turbulent regime will be discussed.

Figure 12 shows the 3D conjugate heat transfer model of the k-ε-optimized flow channel and straight flow channels. The geometry in the XY plane is the same as the design in 3.2 (Figure 2). The height of the flow channel in the Z direction is 10 mm, and the solid base region is 2 mm thick. The bottom of the heat sink is in contact with a heat source, and the boundary condition is uniform heat flux. The inlet and outlet conditions of the fluid are the same as in Section 3.2. The interface between the fluid and solid domain, and the upper and lower walls of the flow channel are given non-slip boundary conditions. Symmetrical boundary conditions are used to simplify the computation process.

The pumping power represents the external energy required to drive the fluid through the cooling channels. Therefore, it is used to analyze the hydraulic characteristics of the heat exchanger. Pumping power is defined as:(21)$$P_{\mathrm{pump}\;}=\Delta P\dot{V}$$
where the $\dot{V}$ is the volume flow of the fluid.

Figure 13 shows the pumping power variation for the two channel designs at different inlet velocities. For both the k-ε-optimized and the straight flow channels, as the flow rate increases, the pumping power consumed by the fluid flow increases rapidly. Compared to the straight flow channel, the k-ε-optimized flow channel consumes less pumping power at the same inlet velocity. This advantage of the k-ε-optimized flow channel is more pronounced when the inlet velocity is higher.

The total thermal resistance and Nusselt number are used to compare the heat transfer performance of the 3D model [34]. The total thermal resistance can be defined as follows:(22)$$R_{\mathrm{th}}=\frac{T_{\mathrm{surf},\;\max\;}-T_{in}}{q}$$
where the $T_{\mathrm{surf},\;\max\;}$ is the maximum temperature of the surface of the conjugate heat transfer model. 

The Nusselt number is defined as:(23)$$Nu=\frac{h_{\mathrm{avg}}\cdot D_{h}}{k_{f}}$$
where the characteristic length D_h_ depends on the cross-sectional area of cooling channel $A_{\mathrm{ch}}$ and the wetted perimeter $C_{w}$ and the relation of them can be described as:(24)$$D_{h}=\frac{4A_{\mathrm{ch}}}{C_{w}}$$
$h_{\mathrm{avg}}$ is the average convective heat transfer coefficient, according to Newton’s law of cooling, it is defined as:(25)$$h_{\mathrm{avg}}=\frac{q}{T_{w,avg}-T_{f,avg}}$$
where $T_{w,avg}$ refers to the average wall temperature. $T_{f,avg}$ is the average fluid temperature and can be calculated as follows:(26)$$T_{f,\;\mathrm{avg}\;}=\frac{T_{\mathrm{in}\;}+T_{\mathrm{out}\;}}{2}$$

Figure 14 shows the variation of the total thermal resistance and Nusselt number of the heat transfer structure under different flow velocities, respectively. The total thermal resistance decreases and the Nusselt number increases with the increase in inlet velocity. When the fluid velocity in a flow channel becomes higher, the thickness of the thermal boundary layer is reduced and the heat transfer between the fluid and the fin is enhanced. For all the inlet velocities used here, the total thermal resistance of the heat sink with the k-ε-optimized flow channel is lower and the Nusselt number is higher than that of the straight flow channel. Therefore, it can be concluded that the 3D k-ε-optimized flow channel improves the convective heat transfer performance of the heat sink compared with the conventional straight channel.

Figure 15 shows the 3D temperature distribution and streamlined distribution of different heat transfer structures with an inlet velocity of 1 m/s. Similar to the results of the 2D models (Figure 7), for the heat sink of the straight flow channel, the abrupt expansion or contraction of the flow channel at the inlet and outlet of the flow channel generates vortices and leads to local kinetic energy loss. On the other hand, for the heat sink with the k-ε-optimized flow channel, the fins evenly distribute the fluid in the flow channels. Moreover, the temperature distribution of the entire heat sink is relatively uniform compared with the straight channels. 

Figure 16 shows the temperature distribution at the bottom of the heat sink in contact with the hot surface. For the heat sink with the straight flow channel, the average temperature is 315.49 K and the maximum temperature is 328.83 K. For the heat sink with the k-ε-optimized flow channel, the average temperature is 314.12 K and the maximum temperature is 317.73 K, indicating a cooler and more uniform temperature profile.

## 4. Conclusions

In summary, we have developed a stable topology optimization scheme for conjugate heat transfer involving single-phase flow in the turbulent regime, and its effectiveness is verified by numerical simulation. From the above discussion, we can draw the following conclusions:It is crucial to add appropriate penalty terms to the turbulent kinetic energy equation and turbulent energy dissipation equation of the k-ε model, respectively. Compared with the Navier–Stokes flow channel and Darcy-optimized flow channel, the k-ε-optimized channel can better balance the relationship between calculation time and design requirements.The heat transfer structure with the k-ε-optimized flow channel has superior hydraulic and heat transfer performance compared to conventional straight channels and Darcy-based TO designs. Elimination of undesirable flow vortices is believed to be an important factor for this performance.A multi-objective function considering the flow resistance as well as the thermal performance is used, and different weight factors are tested. As the objective function puts more weight on heat transfer, the optimized flow channels become narrower and more branched.The k-ε-optimized flow channel has better hydraulic and heat transfer performance in the practical application of the 3D heat sink, which reduces the pumping power and thermal resistance during the heat transfer process and effectively increases the Nusselt number.

Although the article proposes a topology optimization method for turbulent conjugate heat transfer, the method still has some limitations. For example, the variable density method is unable to capture the changes in the solid–liquid interface during the optimization process, and thus the influence of the boundary layer on the surface of solid areas cannot be considered during the optimization process, which affects the accuracy of the topology optimization. We will try to solve this problem in future works.

## Figures and Tables

**Figure 1 entropy-25-01299-f001:**
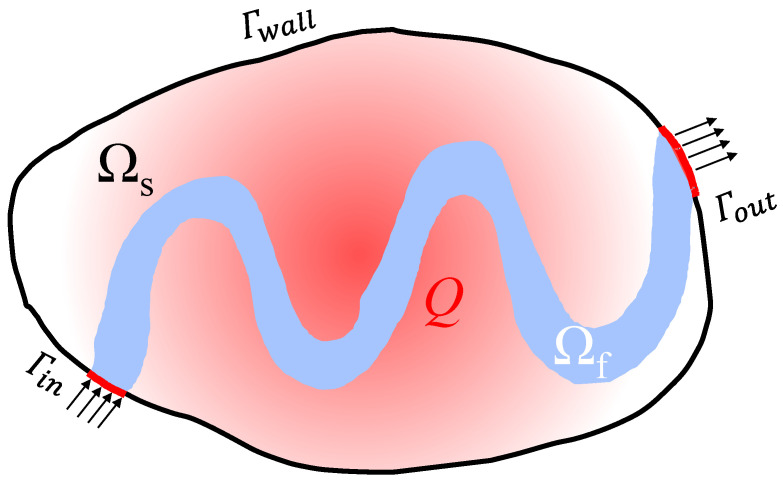
Topology optimization conceptual design model.

**Figure 2 entropy-25-01299-f002:**
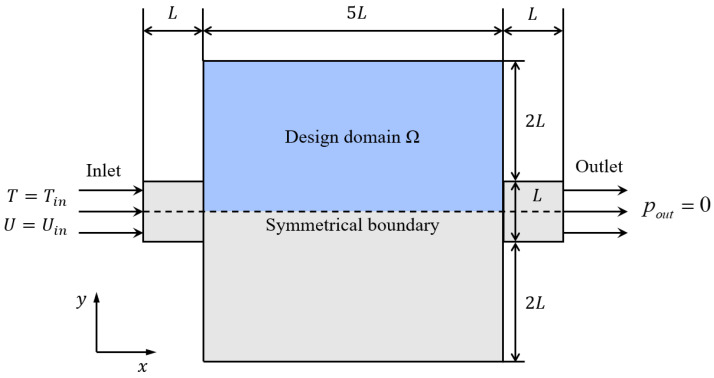
Schematic of the design model and computational settings.

**Figure 3 entropy-25-01299-f003:**
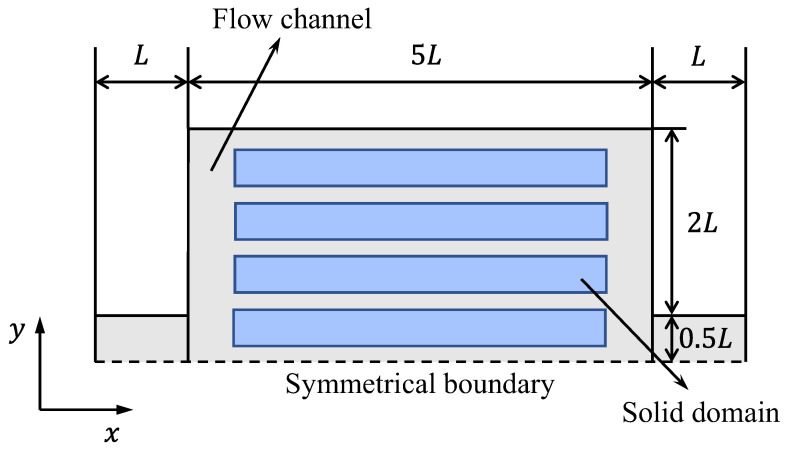
Schematic of traditional straight channels.

**Figure 4 entropy-25-01299-f004:**
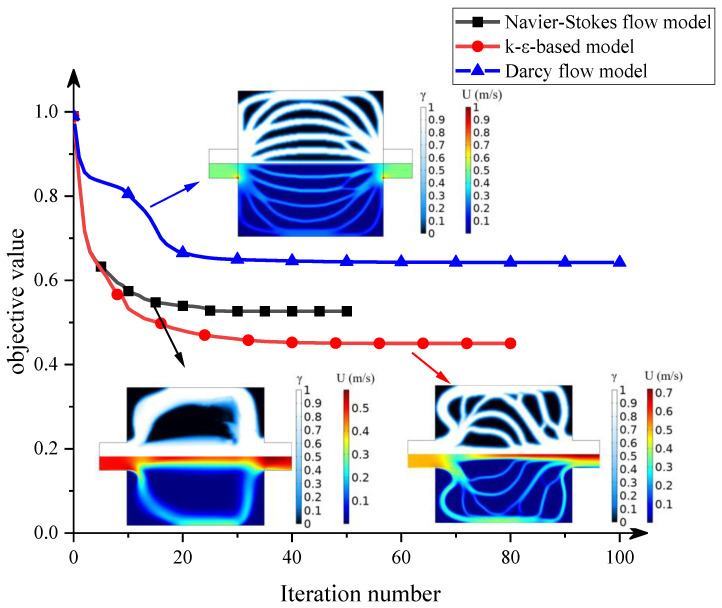
Convergence histories of objective function.

**Figure 5 entropy-25-01299-f005:**
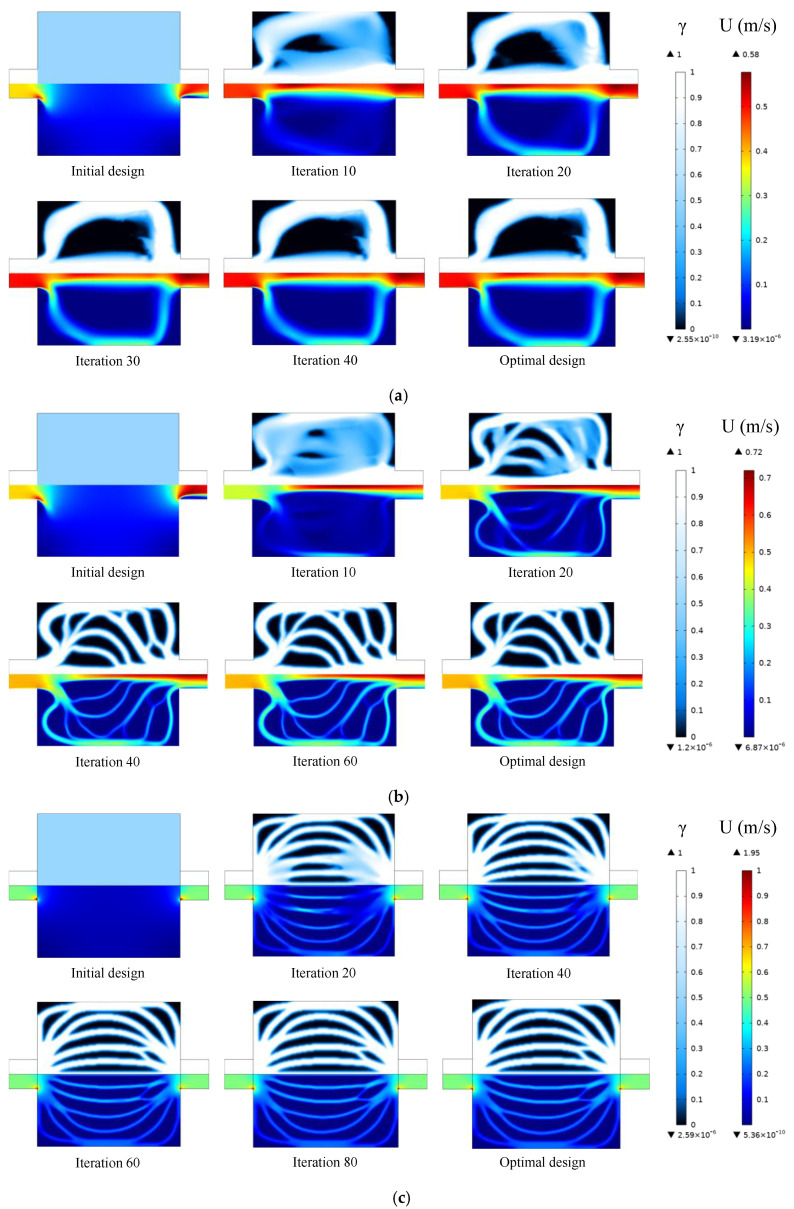
Optimization process for different flow models: (**a**) Navier–Stokes flow model (**b**) k-ε-based model (**c**) Darcy flow model.

**Figure 6 entropy-25-01299-f006:**
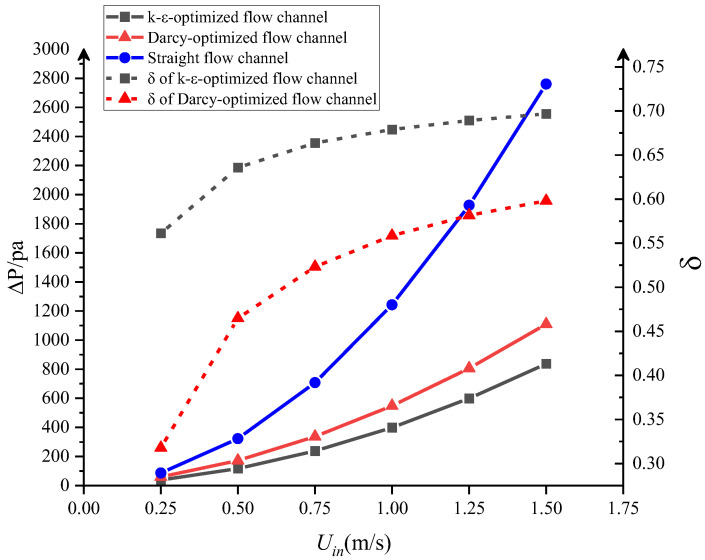
Pressure drop and the relative difference in pressure drop between different flow channel designs.

**Figure 7 entropy-25-01299-f007:**
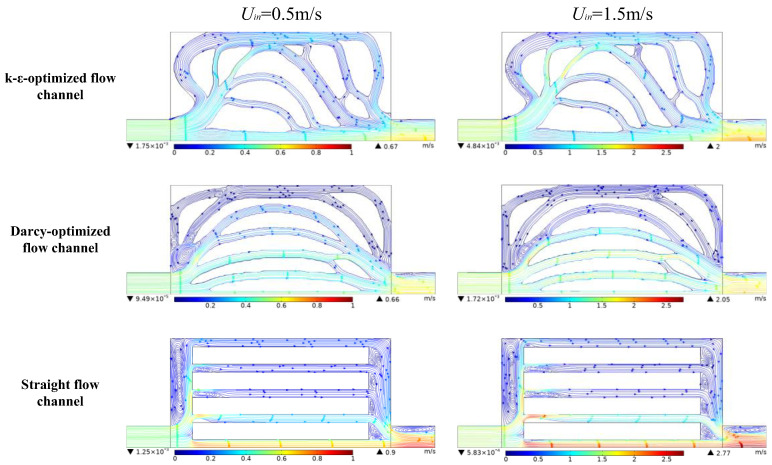
The distribution of streamline in 0.5 m/s and 1.5 m/s.

**Figure 8 entropy-25-01299-f008:**
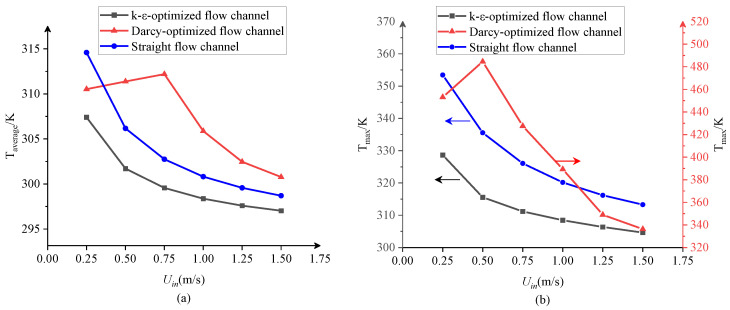
The temperature variation of different flow channels under different inlet velocities. (**a**) Average temperature (**b**) maximum temperature.

**Figure 9 entropy-25-01299-f009:**
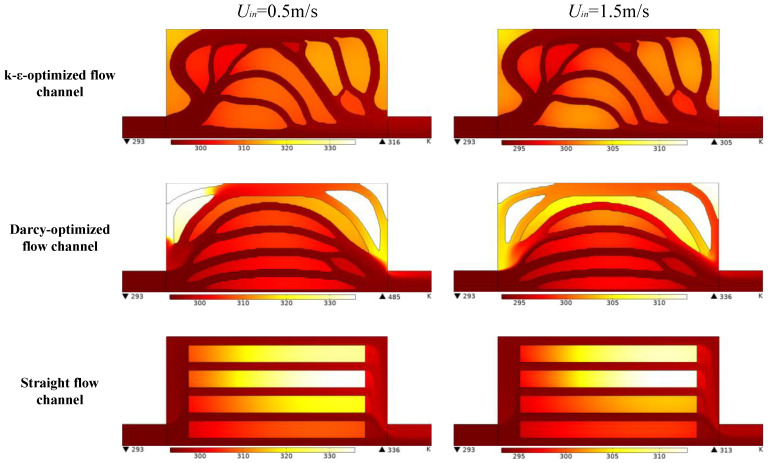
Temperature distribution of different flow channels in 0.5 m/s and 1.5 m/s.

**Figure 10 entropy-25-01299-f010:**
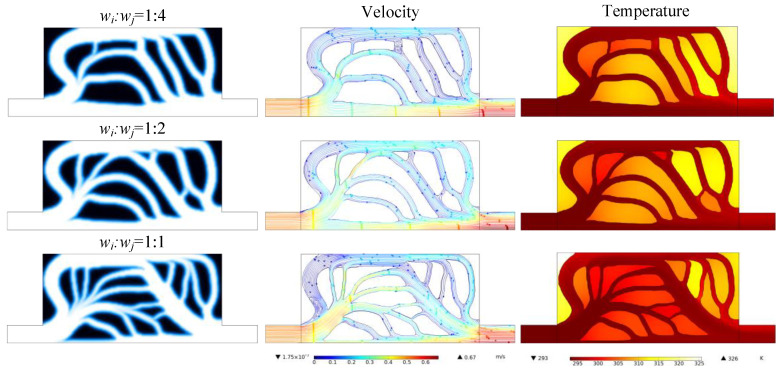
The k-ε-optimized flow channel, the distribution of streamline, and the temperature with different weighting factor ratios *w_i_* (heat transfer):*w_j_* (pressure drop).

**Figure 11 entropy-25-01299-f011:**
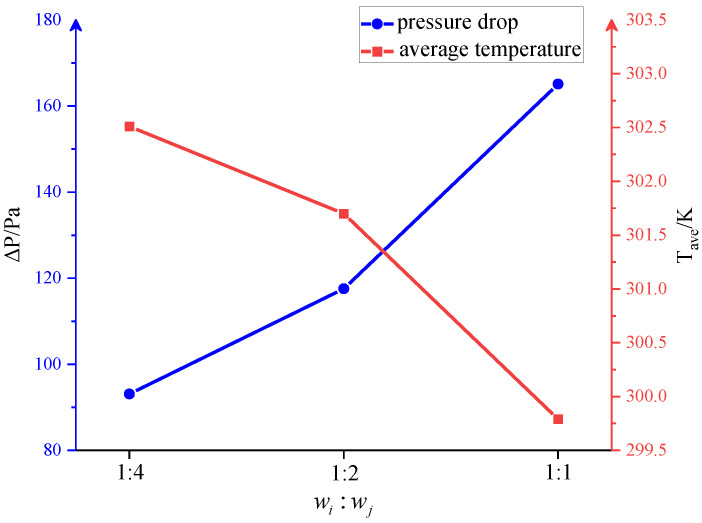
The pressure drop and the average temperature with different weighting factor ratios.

**Figure 12 entropy-25-01299-f012:**
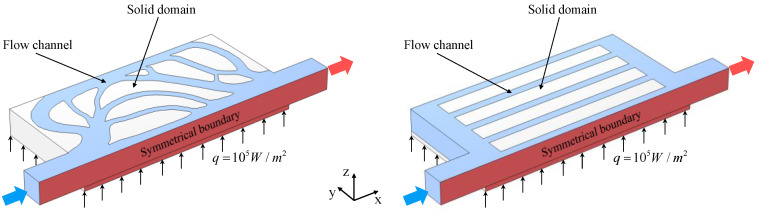
Schematic of 3D conjugate heat transfer model.

**Figure 13 entropy-25-01299-f013:**
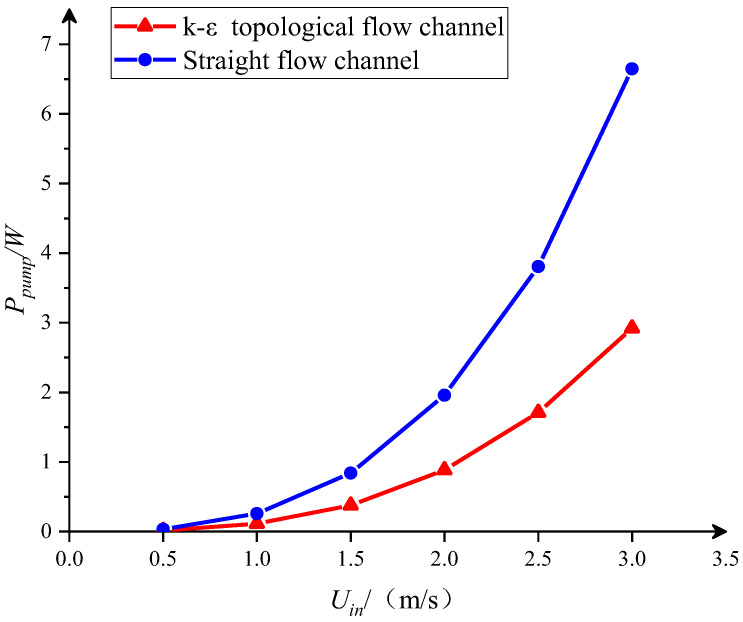
The pumping power consumed by different channel designs.

**Figure 14 entropy-25-01299-f014:**
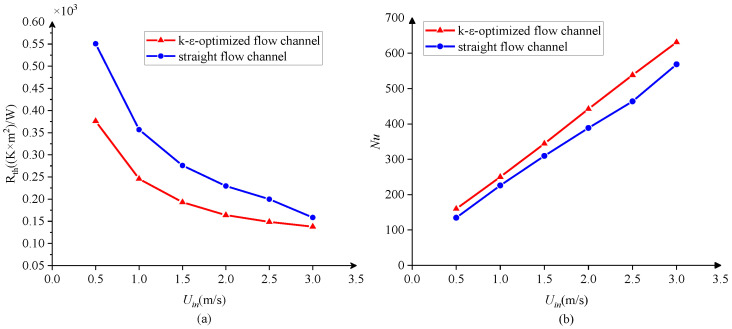
The variation of the total thermal resistance and Nusselt number: (**a**) Total thermal resistance (**b**) Nusselt number.

**Figure 15 entropy-25-01299-f015:**
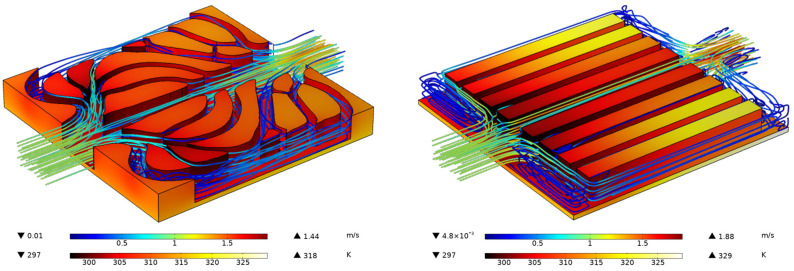
The temperature and velocity distribution of different conjugate heat transfer models.

**Figure 16 entropy-25-01299-f016:**
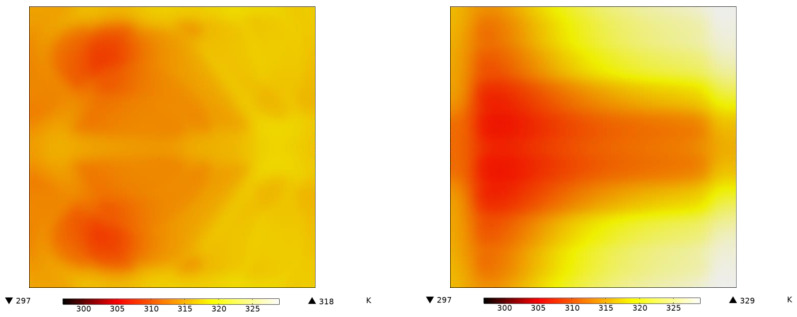
The temperature distribution of the heating surface.

**Table 1 entropy-25-01299-t001:** The values of fluid and solid physical parameters used in this study [21].

$k_{f}$	0.6 W/(m·K)
$k_{s}$	202 W/(m·K)
$\rho_{f}$	1000 kg/m^3^
$\rho_{s}$	2713 kg/m^3^
$c_{f}$	4200 J/(kg·K)
$c_{s}$	871 J/(kg·K)

**Table 2 entropy-25-01299-t002:** The values of computation results.

	Number of Iterations	Computation Time	Average Time of Each Iteration	The Objective Function
Navier–Stokes flow model	50	18,710 s (5 h 11 min 50 s)	374.2 s	0.52705
Darcy flow model	100	202 s (3 min 22 s)	2.02 s	0.64244
k-ε turbulent flow model	80	2590 s (43 min 10 s)	32.375 s	0.45046

**Table 3 entropy-25-01299-t003:** Grid independence test results.

	Grid 1	Grid 2	Grid 3	Deviation: 2–1	Deviation: 2–3
The number of grids	120,780	304,914	516,940		
Average temperature (K)	301.633	301.703	301.711	0.818%	0.094%
Pressure drop (Pa)	119.338	117.551	118.998	1.52%	1.23%

## Data Availability

The data are not publicly available due to privacy.

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
