# Peer review of "Topology Optimization of Turbulent Flow Cooling Structures Based on the k-ε Model"

_entropy, 2023, doi:10.3390/e25091299_

Round 1

Reviewer 1 Report

This paper developed a novel topology optimization involving turbulent conjugated heat transfer. They authors added penalty terms to the Navier-Stokes equations, turbulent kinetic energy equation and turbulent energy dissipation equation. They used interpolation models for the thermal properties of materials. They also developed a multi-objective optimization function to minimize both pressure drop and average temperature. A case study was carried out to comprehensively compare and analyze various optimization methods in turbulent heat transfer.

Overall, this manuscript demonstrates clarity, conciseness, and good writing. It possesses the attributes of a solid journal paper. Thus, I recommend its acceptance.

Reviewer 2 Report

This paper proposes a framework for TO involving turbulent conjugate heat transfer based on the variable density method. Penalty terms are added to the NavierStokes equation, turbulent kinetic energy equation, and turbulent energy dissipation equation, and use interpolation models for the thermal properties of materials. Generally, the research is interesting. There are several issues need to be addressed before the paper can be further considered for publication.

1. The introduction section is a little poor, what is the novelty of the present study related to the model.   

2. A. Gersborg-Hansen[19] is not correct, please revise. Han[21] et al.  should be Han et al. [21],

3. Topology optimization model, how can you integrate it with conjugate heat transfer model.

4. Three models (Navier-Stokes flow model, k- ε-based model, Darcy flow model) are studied and compared, and significant difference are found,which one is the optimal model?

5. As shown in Fig. 8, the results of Darcy optimized flow channel is different, the maximum temperature appears at Um=0.50m/s. Please give the explanation.

6. conclusion should be Conclusion. The conclusion section is not good. The limitations of this study, suggested improvements and future direction of this work should be highlighted; the main findings resulting from the study should be clearly identified.

Slight improvement are needed. 

Round 2

Reviewer 2 Report

The authors addressed all the comments, and the manuscript can be accepted.